# A Guide to Robust Generalization: The Impact of Architecture, Pre-training, and Optimization Strategy

## Abstract

Deep learning models are vulnerable to small input perturbations. For years, robustness to such perturbations was pursued by training models from scratch (i.e., with random initializations) using specialized loss objectives. Recently, robust fine-tuning has emerged as a more efficient alternative: instead of training from scratch, pretrained models are adapted to maximize predictive performance and robustness. To conduct robust fine-tuning, practitioners design an optimization strategy that includes the model update protocol (e.g., full or partial) and the specialized loss objective. Additional design choices include the architecture type and size, and the pretrained representation. These design choices affect robust generalization, which is the model's ability to maintain performance when exposed to new and unseen perturbations at test time. Understanding how these design choices influence generalization remains an open question with significant practical implications. In response, we present an empirical study spanning 6 datasets, 40 pretrained architectures, 2 specialized losses, and 3 adaptation protocols — yielding $1,440$ training configurations and $7,200$ robustness measurements across five perturbation types. To our knowledge, this is the most diverse and comprehensive benchmark of robust fine-tuning to date. While attention-based architectures and robust pretrained representations are increasingly popular, we find that convolutional neural networks pretrained in a supervised manner on large datasets often perform best. Our analysis both confirms and challenges prior design assumptions, highlighting promising research directions and offering practical guidance.

## 1 Introduction

Images processed by machine learning models can contain subtle perturbations that are invisible to the human eye. These perturbations may occur accidentally (e.g. sensor noise, blur, digital format conversions [Jung, 2018]) or intentionally (e.g., adversarial attacks [Szegedy et al., 2014]). Such perturbations can negatively affect the performance of machine learning systems, which is a serious obstacle to their adoption in the real world.

In practice, it is difficult to anticipate which type(s) of perturbation(s) a system may face [Sculley et al., 2015]. A key challenge is therefore to maximize robustness across diverse perturbation types. To achieve that, a typical approach is to assume a set of possible perturbations and induce robustness to this specific set during training [Croce and Hein, 2022, Tramèr and Boneh, 2019, Maini et al., 2020]. However, this strategy is inherently limited, as models may encounter unforeseen perturbations post-deployment [Bashivan et al., 2021, Ibrahim et al., 2022]. In this work, we focus on *robust generalization*: it refers to the ability of models trained for robustness on a specific perturbation type to remain robust to other, unseen, perturbations.

Submitted to 39th Conference on Neural Information Processing Systems (NeurIPS 2025). Do not distribute.

We specifically focus on robust generalization in low data regimes. Robustness-critical applications often face data scarcity constraints, due to data collection costs [Rahimi et al., 2021]. In low data regimes, robust generalization can be induced by fine-tuning for robustness models pre-trained on large datasets [Hua et al., 2024, Xu et al., 2023, Hendrycks et al., 2019, Liu et al., 2023a].

Fine-tuning for robustness involves a wide range of design choices related to the pretrained backbone and the fine-tuning process. When selecting a pretrained backbone, one implicitly selects an architecture type (e.g., convolutional, attention-based, or hybrid), a model size, and a pretraining strategy (e.g., supervised vs. self-supervised, robust vs. non-robust). As for the robust fine-tuning process, one must select a fine-tuning protocol (e.g., partial vs full updates) and a loss objective. A standard loss objective is classic adversarial training (Classic AT) [Madry et al., 2018], which minimizes cross-entropy on adversarially perturbed observations. Another option is the so-called TRADES (TRadeoff-inspired Adversarial DEfense via Surrogate-loss minimization) [Zhang et al., 2019] loss, which optimizes both the cross-entropy and the Kullback-Leibler (KL) divergence between predictions on perturbed and unperturbed observations. Unfortunately, there is currently limited guidance available for practitioners to navigate these choices effectively.

**Research question** *What are the impacts of robust fine-tuning design choices on robust generalization?* Our main hypothesis is that the pretrained backbone interacts with the fine-tuning and optimization strategies to substantially influence robust generalization. Figure 1 motivates this hypothesis by showing important performance variability across design choices and complex interaction patterns among design components.

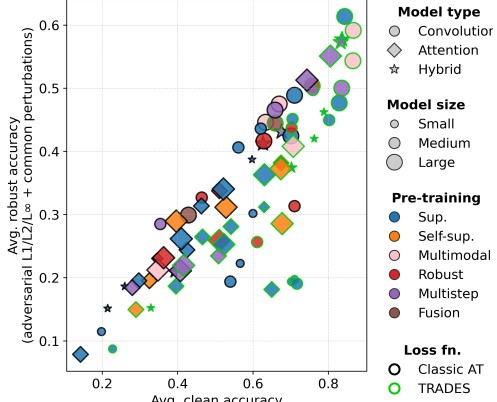

Figure 1: Performance variation across fine-tuning design choices (full fine tuning with 50 epochs). Accuracy averaged over 6 datasets.

**Key findings** We conduct a study on 6 datasets and a total of 240 design choices combinations (40 pretrained backbones × 2 robust losses × 3 fine-tuning protocols). We obtain 7, 200 measurements of robustness on 5 perturbation types. We uncover actionable lessons for practitioners and for future research on robust fine-tuning:

① TRADES loss performs better than Classic AT overall and significantly better in large models. ② Despite growing interest in attention architectures, convolutional architectures show better robust generalization in the considered setups. ③ Hybrid architectures are a promising avenue in robust fine-tuning. ④ With enough compute, supervised pre-training yields best robust generalization, but multi-modal pre-training is also promising. ⑤ Robust pre-training is the clear winner in resource constrained fine-tuning settings. ⑥ When fine-tuning robust backbones with enough compute, using a loss different from the one used for pre-training can boost performance. ⑦ Robust pre-training yields limited returns when scaled to larger architectures. ⑧ Full finetuning is the best overall, and there exist a cost-effective proxy to guide practitioners in finding successful design choices faster.

**Related benchmarks** Tang et al. [2021], Liu et al. [2023b], and Shao et al. [2021] benchmark the performance of different architectures and training strategies on robustness. A main difference is that they all consider "training from scratch" (i.e., training from random initializations). In contrast, our study focuses on fine-tuning from pretrained backbones. Training dynamics observed in one setting do not necessarily transfer to the other [Kornblith et al., 2019]. Another key difference is that the current benchmark analyzes configurations with optimized hyper-parameters (see details in Appendix B), while prior works consider fixed hyper-parameters. This study is therefore better geared towards practitioners. More broadly, this work is inspired by design choices studies in non-robust computer vision [Goldblum et al., 2024] and in robust vision-language [Bhagwatkar et al., 2024].

## 2 Design choices

We study 80 combinations (40 pre-trained backbones × 2 objective losses) using 3 fine-tuning protocols over 6 classification tasks with $C$ classes. Each observation-label pair $(x, y)$ is drawn i.i.d.

from a stationary distribution $(X, Y)$. Each configuration results in a classifier model $f_\theta : X \to \Delta_C$, where $\theta$ are the model parameters and $\Delta_C$ denotes the $(C-1)$-dimensional probability simplex.

## 2.1 Pre-trained backbones

Tremendous progress has been made in the development of pre-trained backbones, and each technique is usually followed by multiple variations. The options available in the open-source community are endless [Wightman, 2019], which motivates an extensive benchmarking of pre-trained backbones.

| Size | Param. Range | Type | Architectures |
|------|--------------|------|---------------|
| Small | 5–10M | Conv. | Regnetx004, Efficientnet-b0, Edgenext (small) |
| | | Attn. | DeiT (tiny) |
| | | Hybrid | Coat (tiny), MobileViT (small) |
| Medium | 25–30M | Conv. | Convnext (tiny), Resnet50 |
| | | Attn. | DeiT (small), ViT (small), Eva02 (tiny), Swin (tiny) |
| | | Hybrid | Coatnet-0 |
| Large | 80–90M | Conv. | Convnext (base) |
| | | Attn. | ViT (base), Eva02 (base), Swin (base) |
| | | Hybrid | Coatnet-2 |

Table 1: Overview of the 19 considered architectures.

**Architectures** We consider a total of 19 architectures, spanning into three size categories: large (80–90 million parameters), medium (25–30 million), and small (5–10 million) – see summary in Table 1. Each architecture is further categorized between one of three structural types: convolutional, attention-based, and hybrid (i.e., mixture of convolution and attention layers). Small architectures are relevant for deployment in low resource environments (e.g., Jetson Nano, Orion) or with low latency requirements. To our knowledge, this is the first study in robust fine-tuning that considers small size architectures (5-10M) [Hua et al., 2024, Xu et al., 2023, Hendrycks et al., 2019, Liu et al., 2023a]. The largest architectures considered are aligned with existing works [Goldblum et al., 2024, Hua et al., 2024]. For larger architectures, we refer to works on scaling robustness [Wang et al., 2024].

**Pre-training protocol** Prior works have studied the influence of supervised pre-training [Hendrycks et al., 2019, Mo et al., 2022], robust pre-training [Hua et al., 2024, Xu et al., 2023, Liu et al., 2023a], and multimodal self-supervised (Multi-SS) pre-training [Hua et al., 2024] in robust fine-tuning. However, these studies are confined to single architecture types and sizes, which restricts the scope of conclusions. In Section 4, we will see that some conclusions do no not hold uniformly across all architecture sizes and types.

| Category | Total | Technical Details |
|----------|-------|-------------------|
| Supervised | 20 | ImageNet-1k/22k; variants with and without data-aug. & regularization |
| Multistep Supervised | 6 | Imagenet-22k then 1k, Imagenet-12k then 1k, variants with and without data-aug. & regularization |
| Robust Supervised | 5 | 4× APGD-K, 1× PGD-K adversarial pre-training; all based on Classic AT on In1k |
| Unimodal Self-Sup. | 4 | MAE, DINO, MIM |
| Multimodal Self-Sup. | 3 | CLIP on LAION-2B / LAION-Aesthetics |
| Fusion | 2 | CLIP (LAION-2B) followed by fine-pass on Imagenet-1k, and Imagenet-12k/1k |

Table 2: Overview of the 40 considered backbones.

Furthermore, this study is the first to compare the performance of pre-training protocols such as supervised (multistep), unimodal self-supervised (Uni-SS), and fusion (i.e., mixture of supervised and self-supervised pre-training) in robust fine-tuning. Understanding how such state-of-the-art pre-training protocols contribute to robust generalization remains a knowledge gap for practitioners.

**Summary** Based on the considered architectures and pre-training protocols, a set of 40 backbones are selected – see summary in Table 2. A global summary of the considered backbones, including exhaustive references and Hugging Face identifiers is available in Appendix A.

## 2.2 Fine-tuning protocols

Consider a pre-trained backbone $g_{\theta_1} : X \to L$, where $L$ denotes an arbitrary latent space. Further consider a classifier $h_{\theta_2} : L \to \Delta_C$ consisting of a linear layer followed by a softmax. The goal of fine-tuning is to combine the pre-trained backbone and the classifier together to obtain a final model $f_\theta : X \to \Delta_C$, with $\theta = \{\theta_1, \theta_2\}$. An observation $x$ is associated to a probability prediction $f_\theta(x) = h_{\theta_2}(g_{\theta_1}(x))$. The fine-tuning process consists of $E$ epochs over the training dataset.

**Full fine-tuning (FFT)** All parameters $\theta = \{\theta_1, \theta_2\}$ are updated for the downstream task. The proposed FFT setup differs from prior works [Hua et al., 2024, Jeddi et al., 2020], who employ a single learning rate across the entire model $f_\theta$. In contrast, our setup allows for distinct learning rates, $\eta_1$ and $\eta_2$, for $g_{\theta_1}$ and $h_{\theta_2}$, respectively, as well as separate weight decay parameters, $\gamma_1$ and $\gamma_2$.

**Linear probing (LP)**    Only the classifier layer $h_{\theta_2}$ is updated, while the parameters of the feature extractor are frozen. The learning rate is $\eta_2$ and the weight decay $\gamma_2$.

We consider three fine-tuning protocols: FFT with $E = 50$ epochs (denoted FFT-50), FFT with $E = 5$ epochs (denoted FFT-5), and LP with $E = 50$ epochs (denoted LP-50). These protocols represent different trade-offs between compute and parameter efficiency. Specifically, LP-50 is parameter-efficient (few trainable weights), while FFT-5 is compute-efficient (short training duration). We do not include LP with 5 epochs as it would combine both constraints and would be very restrictive. Although FFT and LP have been compared before [Hua et al., 2024, Xu et al., 2023, Liu et al., 2023a], there is limited understanding of the compute-efficient setting (FFT-5) and of how design choice combination correlates with performance across the fine-tuning protocols.

**Practical considerations.**    Our choice of 50 training epochs is motivated by prior robust fine-tuning works who employ 40 [Hua et al., 2024] to 60 epochs [Xu et al., 2023, Liu et al., 2023a]. Other (non-robust) fine-tuning benchmarks have considered more epochs (e.g., 100 epochs in Goldblum et al. [2024]) but they are not specifically focused in the low data regime setting. Additional technical details regarding the optimization of hyper-parameters are provided in Appendix B.

## 2.3   Loss objectives

We consider two loss objectives, namely Classic AT and TRADES [Zhang et al., 2019] which are widely popular [Wang et al., 2023, Croce et al., 2020]. There is no consensus as to which loss to choose to perform robust fine-tuning, as suggested by inconsistent design decisions in the literature (e.g., Classic AT in Hua et al. [2024], Singh et al. [2024], TRADES in Xu et al. [2023]). Although Liu et al. [2023b] identify TRADES as most effective, their findings are based on models trained from scratch, which differs from fine-tuning where pre-trained backbones play a central role.

**Crafting synthetic adversarial perturbations**    Both Classic AT and TRADES require crafting synthetic adversarial perturbations throughout training. Given an observation $(x, y)$ and a classifier $f_\theta$, consider the perturbation $x'$ given by the following maximization problem:

$$\arg \max_{x' \in \mathbb{B}(x, \epsilon, p)} \mathcal{L}_{\text{CE}} \left( f_\theta(x'), y \right), \tag{1}$$

where $\mathcal{L}_{\text{CE}}$ denotes the cross-entropy loss and $\mathbb{B}(x, \epsilon, p) = \{ x \in X : \|x' - x\|_p \leq \epsilon \}$ is the $\ell_p$-ball around $x$. Projected Gradient Descent (PGD-$K$) [Madry et al., 2018] with $K$ iterations finds an approximate solution $x'_K$ to the perturbation $x'$ resulting from Eq. 1. Specifically, PGD-$K$ corresponds to starting from $x'_0 = x$ and to iteratively apply the update rule

$$x'_{k+1} = \Pi_{\mathbb{B}(x, \epsilon, p)} \left( x'_k + \delta \, \texttt{sign} \left( \nabla_x \mathcal{L}_{\text{CE}} \left( f_\theta(x'_k), y \right) \right) \right), \quad k = 0, \dots, K - 1 \tag{2}$$

where $\delta \geq 0$ is the step size and $\Pi_{\mathbb{B}(x, \epsilon, p)}$ is the projection operator to ensure that the perturbed input remains within the $\ell_p$-ball. The APGD-$K$ perturbation [Croce and Hein, 2020] improves upon PGD-$K$ by automatically adapting the step size $\delta$, removing the need for manual tuning.

**Classic adversarial training (Classic AT)**    Corresponds to training a classifier $f_\theta(\cdot)$ using the cross-entropy loss $\mathcal{L}_{\text{CE}}$ on observations perturbed by APGD-$K$. This corresponds to minimizing the loss $\mathcal{L}_{\text{AT}}(x, y) := \mathcal{L}_{\text{CE}}(f_\theta(x'_K), y)$ over $\theta$.

**TRADES**    Corresponds to training a classifier $f_\theta(\cdot)$ with the TRADES loss objective [Zhang et al., 2019]. This corresponds to minimizing the following loss over $\theta$:

$$\mathcal{L}_{\text{TRADES}}(x, y) := \mathcal{L}_{\text{CE}}(f_\theta(x), y) + \beta \, \texttt{KL}(f_\theta(x) \| f_\theta(x'_K)), \tag{3}$$

where the scalar $\beta \geq 0$ controls the trade-off between cross-entropy and the Kullback–Leibler (KL) divergence of predictions on perturbed and unperturbed inputs.

**Practical considerations**    To facilitate comparison between Classic AT and TRADES loss objectives, we always consider the same process to craft synthetic adversarial perturbations, namely APGD-$K$ with $K = 10$, $\epsilon = 4/255$, and bounded with respect to the $\ell_\infty$-norm. Additionally, the training data is augmented regardless of the loss objective using standard augmentation techniques (see Appendix B for more details).

# 3 Threat definition and evaluation methods for robust generalization

Although we set a specific type of adversarial perturbation for the optimization strategy (i.e., APGD-K for $\ell_\infty$-norm), deploying machine-learning systems reliably and responsibly requires generalization to diverse, unknown and evolving types of perturbations. We now define additional perturbation types that the model will face at test time, to evaluate robust generalization.

## 3.1 Threat model at test time

To study robust generalization, we define the *threat model* which specifies the possible perturbation types faced by the model at test-time [Akhtar and Mian, 2018]. We use the notation $\mathcal{T}_X(z)$ to denote the distribution of observations drawn from $X$ that contain a perturbation of type $z$. We consider a finite set of perturbation types noted $\tau$, where each type can be categorized into *adversarial* and *common* perturbations.

The adversarial perturbations are bounded by a scalar $\epsilon$ with respect to the $\ell_p$-norm, i.e. $x \sim X, x' \sim \mathcal{T}_X(z)$ such that $\|x - x'\|_p \leq \epsilon$. We include three adversarial perturbation types, generated from $p = 1, 2, \infty$, and $\epsilon = 75.0, 2.0, 4/255$, respectively. The values for $\epsilon$ are standard choices in robustness benchmarks [Croce et al., 2020, Singh et al., 2024]. In contrast, the common perturbations reflect unfortunate events that commonly occur in vision systems (e.g. noise, blur, contrasts, digital format compressions, etc) and that hamper the predictive performance [Jung, 2018].

In summary, the threat model is $\mathcal{T}_X(z), z \in \tau = \{\emptyset, \infty, 1, 2, \texttt{common}\}$, where $\tau$ comprises five perturbation types: no perturbations (i.e., clean observations, noted $\emptyset$), adversarial perturbations under the $\ell_1$, $\ell_2$, and $\ell_\infty$ norm (noted $1, 2$ and $\infty$), and common perturbations (noted $\texttt{common}$). Appendix C describes how we generate these test-time perturbations using open-source software such as AutoAttack [Croce and Hein, 2020] and Jung [2018].

## 3.2 Evaluating robust generalization

We measure performance against the threat model using the *accuracy*, which corresponds to the total number of correct predictions over the total number of observations in the test dataset. We adopt accuracy for its interpretability and widespread use in prior works [Tang et al., 2021], [Liu et al., 2023b], [Shao et al., 2021] and robustness competitions [Croce et al., 2020]. Recall that a configuration is the combination of a pretrained backbone and a loss objective that results into a classifier model. For each of the three fine-tuning protocols considered (FFT-50, FFT-5 and LP-50), we evaluate the performance of each configuration as follows. For every configuration $i \in \{1, \ldots, I\}$ on dataset $d \in \{1, \ldots, D\}$ we obtain a predictive accuracy score $a_{i,d}(z) \in [0, 1]$ for each perturbation type $z \in \tau$. Let $\mathbf{a}_{i,d} := \left[a_{i,d}(z_1), \ldots, a_{i,d}(z_{|\tau|})\right]$ denote the vector of predictive accuracies of configuration $i$ on dataset $d$.

**Borda score** We use the Borda score to compare the relative performance of various configurations on the same fine-tuning protocol. Consider any pair $v = (d, z)$, consisting of a dataset $d$ and a perturbation type $z$ as a voter. Let $V = D \times |\tau|$ be the set of all voters. To each voter $v = (d, z)$ corresponds a function $\texttt{rank}_v : I \to \{1, \ldots, |m_v|\}$ that ranks the configurations $i \in I$ based on their score $a_{i,d}(z)$, in decreasing order. The configuration $i_{\text{top}}$ with best performance gets rank 1 (i.e., $\texttt{rank}_v(i_{\text{top}}) = 1$) and the worst one gets rank $m_v$. We have $m_v \leq |I|$ to account for the possibility of equal scores (and so equal ranks). Then, the Borda score for each configuration $i \in I$ is defined by $B(i) := \sum_{v=1}^{V} m_v - \texttt{rank}_v(i)$.

**Sum score** To account for absolute performance and to compare configurations across different fine-tuning protocols we use the *Sum score*. For each configuration $i \in I$ the sum score is defined by $S(i) := \sum_{(d,z) \in V} a_{i,d}(z)$. By summing the accuracy scores across all perturbation types and datasets, the sum score rewards peak performance even when the accuracy is inconsistent. This contrasts with the Borda score that penalizes inconsistent performance through ranking degradation.

**Mean Absolute Correlation** For each dataset $d$, we define a $|\tau| \times |\tau|$ Spearman correlation matrix $\mathbf{C}^{(J,d)}$ computed over the accuracy vectors $\{\mathbf{a}_{i,d}\}_{j=1}^{J}$ associated to a subset of configurations $J \subseteq I$. The subset $J$ can represent all the configurations ($J = I$) or a subset with a common

specific characteristic (e.g. architecture type, etc). The *mean absolute correlation* for dataset $d$ on the subset of configurations $J$, is noted $\text{MAC}^{(d,J)}$, and is given by $\frac{1}{|\tau|(|\tau|-1)} \sum_{l \neq k} |C_{l,k}^{(J,d)}|$. The $\text{MAC}^{(d,J)}$ is the average absolute off-diagonal correlation between all pairs of perturbation types on dataset $d$ for the subset of configurations in $J$. A high $\text{MAC}^{(d,J)}$, close to 1, indicates that, on average, the performance across all perturbation types is strongly related, suggesting more consistent or uniform robust generalization. Lower $\text{MAC}^{(d,J)}$ values indicate that on average the performance across all perturbation types is less correlated, implying that robustness may be specific to certain perturbations rather than uniform. We also compute a global average across datasets: $\text{MAC}^{(\text{avg},J)} = \frac{1}{D} \sum_{d=1}^{D} \text{MAC}^{(d,J)}$. This informs us on the strength of the robust generalization pattern across datasets.

# 4 Results

We select 6 datasets that fit in the *low data regime* (details in Appendix B). We consider five datasets from the natural image domain (**Caltech101** [Fei-Fei et al., 2004], **Aircraft** [Maji et al., 2013], **Flowers** [Nilsback and Zisserman, 2008], **Oxford pet** [Parkhi et al., 2012], **Stanford cars** [Krause et al., 2013] ) and one from the satellite imagery domain (**Land-Use** [Yang and Newsam, 2010]).

We benchmark 240 design combinations (40 pretrained backbones × 2 robust losses × 3 finetuning protocols) over 6 datasets, totaling 1,440 evaluated configurations.

The hyper-parameters of each configuration are independently optimized (details are reported in Appendix B). Each configuration is tested against 5 perturbation types, unseen during (pre-) training (Section 3.1), resulting in 7,200 robustness measurements. To our knowledge, this benchmark includes the most diverse and comprehensive set of design choices in the robust fine-tuning setting. Collected measurements, and code are open-sourced[1].

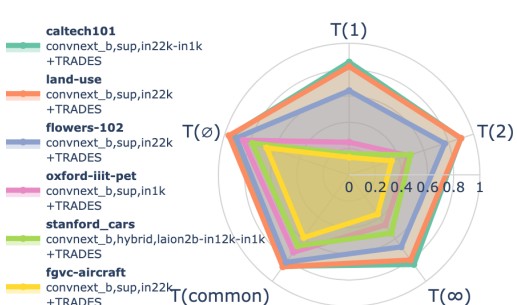

Figure 2: Robust generalization of the best configuration per dataset using FFT-50 (Borda score).

## 4.1 Which configurations perform best?

**Best performing configurations overall.** Table 3 reports the best performing configurations in FFT-50. We see that the best performing backbone is convolutional (*Convnext (base), with supervised pretraining on Imagenet-22k, using TRADES*). FFT-50 clearly outperforms other fine-tuning protocols, with a best sum score of 19.79, which is 61% higher than FFT-5 and 53% higher than LP-50 (see Table 8 and 10 in Appendix). In the Appendix, Tables 7, 9, and 11 report the ranking of all the configurations across FFT-50, FFT-5, and LP-50 respectively.

**Best performing configurations per dataset.** Figure 2 displays the best performing configuration per dataset, when using FFT-50. We observe that convolutional architectures outperform other options on all considered datasets. Despite growing attention on the robustness of attention-based architectures [Bai et al., 2021, Liu et al., 2023b, Shao et al., 2021], our findings show that the robust generalization capacity of well tuned convolutional architectures should not be underestimated. Additionally, on two datasets (Caltech101, and Land-Use), the best configurations achieve accuracy above 0.8 on all perturbation types, demonstrating strong robust generalization. This performance is remarkably high for the field [Croce et al., 2020], demonstrating the practical potential of robust fine-tuning and the importance of carefully identifying best design choices.

**Low-cost proxies exist in robust fine-tuning.** Given the evolving set of available design choices, practitioners need to be equipped with low-cost tools to rapidly identify design choices that are more promising than others. The identification of low-cost proxies helped practitioners in natural language

---

[1] https://anonymous.4open.science/r/robust_training-636C/README.md

modeling [Zhu et al., 2022], and can also benefit costly protocols such as robust learning. Between LP-50 and FFT-5, we find that LP-50 is the most reliable low-cost proxy to FFT-50 (see Figure 8a), especially when using TRADES over Classic AT. Indeed, the correlation between LP-50 and FFT-50 using TRADES is the highest.

| Size | Gold (1st) | Silver (2nd) | Bronze (3rd) |
|---|---|---|---|
| small | coat_t,sup,in1k, TRADES (GR:18, BS:1653, SS:15.83) | edgenetx_s,sup,in1k, TRADES (GR:23, BS:1552, SS:14.66) | edgenetx_s,sup,in1k, Classic AT (GR:33, BS:1356, SS:12.88) |
| medium | convnext_t,sup,in22k-in1k, TRADES (GR:14, BS:1773, SS:16.49) | convnext_t,sup,in1k, TRADES (GR:15, BS:1681, SS:15.6) | convnext_t,sup,in22k, TRADES (GR:20, BS:1650, SS:15.07) |
| large | convnext_b,sup,in22k, TRADES (GR:1, BS:2281, SS:19.79) | coatnet_2,sup,in12k-in1k, TRADES (GR:2, BS:2127, SS:18.74) | coatnet_2,sup,in12k, TRADES (GR:3, BS:2116, SS:18.87) |

Table 3: Top FFT-50 configurations, with global ranking (GR) based on Borda score (BS), sum score (SS) also reported below.

## 4.2 Design Choices Favoring TRADES in Robust Fine-Tuning.

**Overall, TRADES outperforms Classic AT.** It has been shown previously that TRADES outperforms Classic AT when training from scratch [Liu et al., 2023b]. Our results show that these conclusions hold in the fine tuning setting as well (see Figure 4 in the Appendix). We next extend these results by identifying strong interactions between the loss and other design choices in the FFT-50 setting (see Figure 3). The identification of such interactions with TRADES is particularly valuable, given its frequent association with state-of-the-art performance on robustness benchmarks [Wang et al., 2023, Croce et al., 2020].

**TRADES interacts positively with architecture size.** On average, TRADES achieves higher returns compared to Classic AT when architecture size grows (see Figure 3a). Additionally, the odds ratio of TRADES outperforming Classic AT increases steeply with architecture scale, which is a significant effect in FFT-50 (see Figure 5 in Appendix E). These results suggest that TRADES is a promising approach to improve the robustness of large systems, a setting where Classic AT is currently the preferred approach [Wang et al., 2024]. Existing implementations of TRADES require the storage of two forward passes in memory, which motivates an avenue to improve this algorithmic limitation to fully reveal the potential of TRADES on large architectures.

**TRADES interacts best with convolutional and hybrid architectures.** While TRADES and Classic AT yield equivalent outcomes (similar mean Borda score) for attention-based architectures, convolutional and hybrid architectures benefit most from using TRADES over Classic AT (see Figure 3b). Since convolutional architectures tend to overfit more local features and patterns [Bhojanapalli et al., 2021], this result suggests that TRADES regularizes convolutional architectures more efficiently than Classic AT in computer vision tasks.

## 4.3 Distinct robust generalization patterns across architectures sizes and types.

**Larger architectures are consistently better.** Large architectures clearly outperform medium and small architectures in FFT-50 (see Table 3) and generalize better (see Table 7). Large convolutional and hybrid architectures outperform attention-based ones on average (see Figure 3d), though attention models may show their full potential at larger scales [Wang et al., 2024]. Because model scale is often subject to limitations in practice, we also provide analysis at specific architecture sizes to guide practitioners with such limitations.

**If constrained to small architectures, hybrid architectures are the best option.** Among small architectures, hybrid architectures achieve significantly higher scores than fully convolutional ones (see Figure 3d). Using TRADES loss, *Coat (tiny)* and *EdgeNetx* rival larger architectures and achieve impressive rankings of GR:18 and GR:23, corresponding to tier-1 performance among 80 configurations (see Table 3). Prior works on robustness are generally focused on larger architectures [Liu et al., 2023b], but this result extends knowledge by highlighting the practical potential of hybrid architectures for robust fine-tuning using small architectures on data scarce regimes. The result also

constitutes valuable motivations for the community that supports hybrid architectures [Dai et al., 2021, Maaz et al., 2022, Dai et al., 2021].

**Promising generalization properties of hybrid architectures.** Table 4 shows that hybrid architectures achieve the highest MAC values compared to attention and convolutional architectures in FFT-50. This finding complements prior work demonstrating strong robustness of hybrid architectures trained from scratch [Liu et al., 2023b], extending these conclusions to the robust fine-tuning setting. We also generalize the observation across diverse scales and types of hybrid architectures, while the previous observation held only for CoatNet (16M). Finally, Table 4 provides a precise characterization on the robust generalization capability of hybrid architectures, beyond accuracy score.

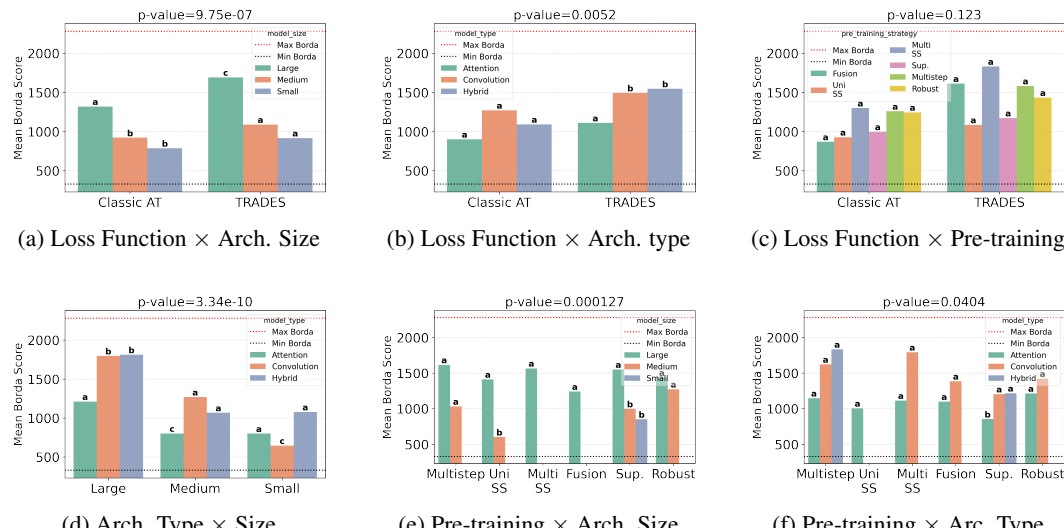

(a) Loss Function × Arch. Size    (b) Loss Function × Arch. type    (c) Loss Function × Pre-training

(d) Arch. Type × Size    (e) Pre-training × Arch. Size    (f) Pre-training × Arc. Type

Figure 3: Nested Welch's ANOVA of the form A × B testing the main effect of A and how the effect of B varies within each level of A (p-value on top). Post-hoc groupings from Tukey HSD tests are annotated with letters above the bars: bars with different letters belong to significantly different groups, at the 90% confidence level. Results for FFT-50.

## 4.4 Influence of the pre-training strategy on robust generalization

**Multi-modal self-supervised pretraining is beneficial for convolutional architectures.** In FFT-50, the *Convnext (base)* architecture with multimodal self-supervised pre-training using the TRADES loss achieves the fourth best ranking in terms of Borda score, and top-2 in terms of sum score (see Table 7 in Appendix E). This performance is surprising given prior results [Hua et al., 2024] showing that supervised and robust pre-training often outperform Multi-SS. However, prior works focus on Multi-SS of attention-based architectures, while the reported performance improvement targets Multi-SS on convolutional architectures (see Figure 3e). The potential of Multi-SS pre-training is further evidenced in Figure 3c, where Fusion (which is based on Multi-SS) pre-training achieves second best average performance, behind Multi-SS.

**Robust pre-training performs best in constrained fine-tuning protocols.** In FFT-5, the global gold (global rank GR:1), silver (GR:2), and bronze (GR:3) are achieved with robust pretraining (see Table 8 in Appendix E). Similarly, the top-3 in LP-50 (GR:1,2 and 3) are also achieved with robust pre-training (see Table 10 in Appendix E). Note that this competitive performance does not hold in the less constrained FFT-50 protocol, where the best configuration based on robust pre-training achieves a global ranking of 9 (see Table 7). Our results are aligned with prior results showing that robust pre-training helps in parameter-efficient settings such as low-rank adaptation [Xu et al., 2023, Liu et al., 2023a] and linear probing [Hua et al., 2024]. We further show that robust pre-training is also beneficial in fine-tuning protocols constrained on the number of updates, a setting not covered by prior works.

**Robust pre-training of larger architectures (may) have limited returns.** Given its high computational cost, robust pre-training prompts critical evaluation of its return on investment relative to alternative pre-training protocols. The performance gains from medium to large architectures are relatively modest for robust pre-training (see Figure 3e). The rate of improvement [2] is $+12\%$ for robust pre-training, whereas it reaches $+43\%$ for supervised, $+44\%$ for multi-step, and $+80\%$ for Uni-SS in FFT-50. This relatively low gain is due to already high performance of robust pre-training at the medium scale, leaving less room for improvement at the larger scale. Although recent works have emphasized scaling robust pretraining to large architectures Singh et al. [2024], there are currently no robust pre-trained small architectures. This finding suggests that robust pre-training at smaller architecture scales could be a promising and underexplored direction for future research.

**Influence of loss objective switches between robust pre-training and robust fine-tuning.** When specifically considering robust pre-trained architectures, a question for practitioners is: *should we use the same robust loss objective for fine-tuning as for pre-training?* With FFT-50, configurations that use a different loss objective for robust fine-tuning than for robust pre-training significantly outperform configurations that use the same loss for both phases. While the global mean between both choices are not statistically different (e.g., $p = 0.42$ with the Mann–Whitney test), we observe that switched configurations are strongly overrepresented among top performers across the 6 datasets considered: 5 out of 6 top-1 configurations use a loss switch, with a binomial $p$-value of $0.041 < 0.05$. Previous works have used switching [Xu et al., 2023, Liu et al., 2023a] and non-switching strategies [Hua et al., 2024]. Our finding provides the first evidence that switching losses between pre-training and fine-tuning can be beneficial with enough compute (result holds only in FFT-50).

| | | Caltech101 | Aircraft | Flowers-102 | Oxford-pet | Stanford-cars | Land-use | Global $\text{MAC}_{\text{avg}}$ |
|---|---|---|---|---|---|---|---|---|
| MAC per dataset $\text{MAC}_d$ over the 80 configs. | | 0.847 | 0.782 | 0.805 | 0.681 | 0.849 | 0.807 | 0.795 |
| Loss objective | Classic AT | 0.876 | 0.702 | 0.842 | 0.778 | 0.844 | 0.890 | 0.822 |
| | TRADES | 0.823 | 0.884 | 0.830 | 0.669 | 0.896 | 0.777 | 0.813 |
| Architecture size | Large | 0.825 | 0.882 | 0.860 | 0.859 | 0.871 | 0.803 | 0.850 |
| | Medium | 0.743 | 0.628 | 0.743 | 0.436 | 0.701 | 0.833 | 0.681 |
| | Small | 0.911 | 0.531 | 0.503 | 0.467 | 0.575 | 0.678 | 0.611 |
| Architecture type | Attention | 0.884 | 0.800 | 0.761 | 0.713 | 0.856 | 0.882 | 0.816 |
| | Convolutional | 0.762 | 0.756 | 0.811 | 0.574 | 0.818 | 0.719 | 0.740 |
| | Hybrid | 0.951 | 0.815 | 0.892 | 0.798 | 0.889 | 0.879 | 0.871 |
| Pre-training protocol | Fusion | 0.760 | 0.760 | 0.920 | 0.660 | 0.920 | 1.000 | 0.837 |
| | Uni-SS | 0.821 | 0.902 | 0.860 | 0.619 | 0.878 | 0.668 | 0.791 |
| | Multi-SS | 0.911 | 0.742 | 0.760 | 0.589 | 0.855 | 0.703 | 0.760 |
| | Supervised | 0.937 | 0.760 | 0.727 | 0.682 | 0.806 | 0.789 | 0.783 |
| | Multistep | 0.867 | 0.809 | 0.846 | 0.893 | 0.856 | 0.798 | 0.845 |
| | Robust | 0.568 | 0.641 | 0.841 | 0.670 | 0.685 | 0.877 | 0.714 |

Table 4: Summary table of the Mean Absolute Correlation in FFT-50, measured over the 80 configurations as well as on subsets of configurations based on design choices.

# 5 Conclusion

Among other findings, we find that convolutional architectures perform best for robust fine-tuning in the low-data regime. Despite growing interest in the robustness of attention-based architectures [Bhojanapalli et al., 2021], our study suggests they are more difficult to fine-tune for robustness in practice. Our findings have broader design impacts for vision systems: for example, vision-language models predominantly rely on attention-based backbones [Radford et al., 2021].

**Limitations** The insights from this study are contingent to the set of datasets, backbones, and optimization strategies considered. We acknowledge that such insights need to continually evolve with the development of new design choices. In this study, the configurations were optimized based on a total compute budget, rather than on an equal number of trials across backbones. This choice reflects the practical reality that some backbones are more challenging to tune due to their compute requirements. This compute-aware tuning approach reflects real-world deployment constraints and promotes energy-conscious model selection [Courty et al., 2024].

---

[2]Rate measured using the relative change w.r.t. the average of the two scores, and computed as $\frac{\text{large}-\text{medium}}{(\text{large}+\text{medium})/2}$. This rate to ensures a symmetric and unbiased comparison that does not privilege either model scale.

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
