# OpenReview forum: "A Guide to Robust Generalization: The Impact of Architecture, Pre-training, and Optimization Strategy"
_NeurIPS.cc/2025/Workshop/Reliable_ML — NeurIPS 2025 - Reliable ML Workshop_

### Official Review · Reviewer_nxJR · 2025-09-18
**The authors examine the role of robust fine-tuning in enabling models to generalize under unseen perturbations—a concept known as robust generalization. They design an experimental study to investigate the importance of different parameters for the robust generalization of fine-tuned models.**

**Rating:** 7
**Confidence:** 2

**Review:**

This paper presents an empirical study examining how different design choices (architecture type, model size, pretrained representations, update protocols, and loss objectives) affect the robust generalization of fine-tuned deep learning models against adversarial perturbations. The study, spanning 6 datasets and over 1,400 training configurations, reveals that convolutional neural networks pretrained on large datasets often outperform popular attention-based architectures for robust fine-tuning, challenging common assumptions in the field.

Strengths:

1.Comprehensive evaluations across multiple dimensions: datasets, pre-trained models, loss objectives, fine-tuning protocols, and classification tasks.

Weaknesses/Suggestions:
While I am not deeply familiar with this research area, I have the following questions:
1.Does the original training algorithm used for the pre-trained model impact robust generalization after fine-tuning? Is there an inherent trade-off between standard accuracy and robustness?

2.What is the rationale behind the chosen values of ε for the perturbation magnitude? Why not experiment with a broader range of ε values to better understand the robustness-perturbation relationship?